# Top-Quality: Finding Practically Useful Sets of Best Plans

**Michael Katz** and **Shirin Sohrabi** and **Octavian Udrea**

IBM T.J. Watson Research Center

1101 Kitchawan Rd, Yorktown Heights, NY 10598, USA

## Abstract

The need for finding a set of plans rather than one has been motivated by a variety of planning applications. The problem is studied in the context of both diverse and top-k planning: while diverse planning focuses on the difference between pairs of plans, the focus of top-k planning is on the quality of each individual plan. Recent work in diverse planning introduced additionally restrictions on solution quality. Naturally, there are application domains where diversity plays the major role and domains where quality is the predominant feature. In both cases, however, the amount of produced plans is somewhat an artificial constraint, and the actual number has little meaning.

Inspired by the recent work in diverse planning, we propose a new computational problem called *top-quality* planning, where solution validity is defined through plan quality bound rather than an arbitrary number of plans. Switching to bounding plan quality allows us to implicitly represent sets of plans. In particular, it makes it possible to represent sets of valid plan reorderings with one plan. We formally define the corresponding computational problem and present the first planner for that problem. We empirically demonstrate the superior performance of our approach compared to a top-k planner-based baseline, ranging from $49\%$ increase in coverage for finding all optimal plans to $69\%$ increase in coverage for finding all plans of quality up to $120\%$ of optimal plan cost.

## 1 Introduction

While the main focus in classical planning was on producing a single plan, a variety of applications has shown the need for finding a set of plans rather than one. These applications include malware detection (Boddy et al. 2005), plan recognition as planning and its applications (Riabov et al. 2015; Sohrabi, Riabov, and Udrea 2016; Sohrabi et al. 2018; Shvo, Sohrabi, and McIlraith 2018), human team planning (Kim et al. 2018), explainable AI (Chakraborti et al. 2018), re-planning and plan monitoring (Fox et al. 2006).

The problem of finding a set of plans is studied in the context of both diverse planning (e.g., Nguyen et al. 2012) and top-k planning (e.g., Katz et al. 2018). Diverse planning focuses on the difference between pairs of plans, evaluating a set of plans by aggregating over the pairwise differences between plans in the set. Recent work in diverse planning introduced additional restrictions on solution quality, requiring each plan in the set to also be of bounded quality (Vad-

lamudi and Kambhampati 2016; Katz and Sohrabi 2019). Top-k planning is a generalization of cost-optimal planning. The focus of top-k planning is on the quality of each individual plan, guaranteeing that no plan of better cost exists outside the solution of a requested size.

Naturally, there are application domains where diversity plays the major role and domains where quality is the predominant feature. The latter include plan recognition (Sohrabi, Riabov, and Udrea 2016), multi-agent plan recognition (Shvo, Sohrabi, and McIlraith 2018), human team planning (Kim et al. 2018), and explainable AI (Chakraborti et al. 2018). These applications exploit top-k planners to derive a large number of plans. In these domains, though, the focus on the number of plans provided is somewhat artificial, and is intended solely to ensure that a sufficient number of plans is found. Further, ordering of actions in a plan can be of less importance in some applications. Plan recognition is one such example application. In plan recognition as planning (Sohrabi, Riabov, and Udrea 2016; Shvo, Sohrabi, and McIlraith 2018), a planning task consists of actions that explain/discard observations. There is no meaning to the order among these actions. Some specific practical applications for plan recognition are hypothesis generation (Sohrabi et al. 2016) and scenario planning advisor (Sohrabi et al. 2018). These applications use a top-k planner with a large bound on the number of required plans $k$, and the obtained plans are post-processed to discard reorderings and cluster similar plans. This would also apply to e.g., problems with actions that correspond to information gathering, where no particular ordering is required. The clear disadvantage of a top-k planner in such cases is that it would generate all possible orderings before proceeding to plans of a higher cost. Thus, the number of required plans used in practice is often a crude over-approximation. Further, even quite large numbers are often not sufficient to ensure that the set of plans includes enough plans of interest, since plans can easily have millions of valid reorderings. A top-k planner would have to generate all these plans before it can get to a plan of a higher cost. Diverse planners (Bryce 2014; Nguyen et al. 2012; Coman and Muñoz-Avila 2011; Roberts, Howe, and Ray 2014; Vadlamudi and Kambhampati 2016) tackle the issue by defining diversity criteria over a set of plans, but only a handful of works take the plan quality into consideration (Roberts, Howe, and Ray 2014; Vad-

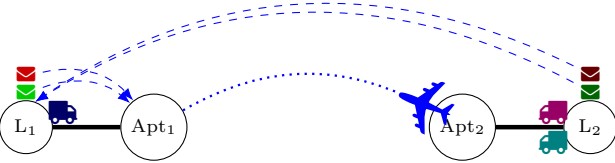

Figure 1: Example logistics task.

lamudi and Kambhampati 2016; Katz and Sohrabi 2019). While some computational problems in diverse planning do require to provide some guarantees on solution quality (Katz and Sohrabi 2019), existing diverse planners still do not provide such guarantees.

In this paper, we propose a new family of computational problems called *top-quality planning*. The objective of top-quality planning is to find and concisely represent a set of all plans of bounded quality, for a given (absolute) bound. That is, we suggest an alternative definition of solution validity, by bounding the solution quality instead of bounding the number of plans. This allows us to define an *equivalence* relation on plans and implicitly represent equivalence classes plans without knowing the exact number of plans in the set. In particular, in this work, we focus on the equivalence relation that is defined by all possible reorderings of each plan, represented by one *canonical* plan. Furthermore, we propose a first planner for *unordered* top-quality planning that iteratively finds a single plan of top quality and forbids at once all plans found so far, including all their possible reorderings. For that, we adapt a recently proposed diverse planning reformulation that forbids a single multiset of actions (Katz and Sohrabi 2019) to forbid exactly a collection of multisets. Our adaptation of the existing reformulation allows us to forbid multiple sets of plans at each iteration while preserving soundness and completeness of our approach. We empirically compare our approach to unordered top-quality planning to the only available baseline – a top-k planner with a large k bound. Our approach exhibits a superior performance, ranging from 49% increase in coverage for finding all optimal plans to 69% increase in coverage for finding all plans of cost up to 120% of optimal plan cost.

## 2 Preliminaries

We consider classical planning tasks in the well-known SAS$^+$ formalism (Bäckström and Nebel 1995), extended with action costs. Such *planning tasks* $\Pi = \langle \mathcal{V}, \mathcal{O}, s_0, s_\star \rangle$ consist of $\mathcal{V}$, a finite set of finite-domain *state variables*, $\mathcal{O}$, a finite set of *actions*, $s_0$, an *initial state*, and $s_\star$, the *goal*. Each variable $v \in \mathcal{V}$ is associated with a finite domain $dom(v)$ of variable values. These variable, value pairs are called *facts*. A *partial assignment* $p$ maps a subset of variables $vars(p) \subseteq \mathcal{V}$ to values in their domains. For a variable $v \in \mathcal{V}$ and partial assignment $p$, the value of $v$ in $p$ is denoted by $p[v]$ if $v \in vars(p)$ and we say $p[v]$ is *undefined* if $v \notin vars(p)$. A partial assignment $s$ with $vars(s) = \mathcal{V}$, is called a *state*. State $s$ is *consistent* with partial assignment $p$ if they agree on all variables in $vars(p)$, shortly denoted by $p \subseteq s$. The product $\mathcal{S} = \prod_{v \in \mathcal{V}} dom(v)$ is called the

| $\pi_a$ | $\pi_b$ | $\pi_c$ |
|---|---|---|
| (load P$_4$ T$_2$ L$_2$) | (load P$_3$ T$_2$ L$_2$) | (load P$_4$ T$_3$ L$_2$) |
| (load P$_3$ T$_2$ L$_2$) | (load P$_4$ T$_2$ L$_2$) | (load P$_3$ T$_3$ L$_2$) |
| (drive T$_2$ L$_2$ Apt$_2$) | (drive T$_2$ L$_2$ Apt$_2$) | (drive T$_3$ L$_2$ Apt$_2$) |
| (unload P$_4$ T$_2$ Apt$_2$) | (unload P$_4$ T$_2$ Apt$_2$) | (unload P$_4$ T$_3$ Apt$_2$) |
| (unload P$_3$ T$_2$ Apt$_2$) | (unload P$_3$ T$_2$ Apt$_2$) | (unload P$_3$ T$_3$ Apt$_2$) |
| (load P$_2$ T$_1$ L$_1$) | (load P$_2$ T$_1$ L$_1$) | (load P$_2$ T$_1$ L$_1$) |
| (load P$_1$ T$_1$ L$_1$) | (load P$_1$ T$_1$ L$_1$) | (load P$_1$ T$_1$ L$_1$) |
| (load P$_3$ A Apt$_2$) | (load P$_3$ A Apt$_2$) | (load P$_3$ A Apt$_2$) |
| (load P$_4$ A Apt$_2$) | (load P$_4$ A Apt$_2$) | (load P$_4$ A Apt$_2$) |
| (fly A Apt$_2$ Apt$_1$) | (fly A Apt$_2$ Apt$_1$) | (fly A Apt$_2$ Apt$_1$) |
| (unload P$_3$ A Apt$_1$) | (unload P$_3$ A Apt$_1$) | (unload P$_3$ A Apt$_1$) |
| (unload P$_4$ A Apt$_1$) | (unload P$_4$ A Apt$_1$) | (unload P$_4$ A Apt$_1$) |
| (drive T$_1$ L$_1$ Apt$_1$) | (drive T$_1$ L$_1$ Apt$_1$) | (drive T$_1$ L$_1$ Apt$_1$) |
| (load P$_4$ T$_1$ Apt$_1$) | (load P$_4$ T$_1$ Apt$_1$) | (load P$_4$ T$_1$ Apt$_1$) |
| (load P$_3$ T$_1$ Apt$_1$) | (load P$_3$ T$_1$ Apt$_1$) | (load P$_3$ T$_1$ Apt$_1$) |
| (unload P$_2$ T$_1$ Apt$_1$) | (unload P$_2$ T$_1$ Apt$_1$) | (unload P$_2$ T$_1$ Apt$_1$) |
| (unload P$_1$ T$_1$ Apt$_1$) | (unload P$_1$ T$_1$ Apt$_1$) | (unload P$_1$ T$_1$ Apt$_1$) |
| (drive T$_1$ Apt$_1$ L$_1$) | (drive T$_1$ Apt$_1$ L$_1$) | (drive T$_1$ Apt$_1$ L$_1$) |
| (unload P$_4$ T$_1$ L$_1$) | (unload P$_4$ T$_1$ L$_1$) | (unload P$_4$ T$_1$ L$_1$) |
| (unload P$_3$ T$_1$ L$_1$) | (unload P$_3$ T$_1$ L$_1$) | (unload P$_3$ T$_1$ L$_1$) |

Figure 2: Three example cost-optimal plans for the example task.

*state space* of planning task $\Pi$. $s_0$ is a state and $s_\star$ is a partial assignment. A state $s$ is called a *goal state* if $s_\star \subseteq s$ and the set of all goal states is denoted by $\mathcal{S}_{s_\star}$. Each action $o$ in $\mathcal{O}$ is a pair $\langle pre(o), eff(o) \rangle$ where $pre(o)$ is a partial assignment called *precondition* and $eff(o)$ is a partial assignment called *effect*. Further, $o$ has an associated natural number $cost(o)$, called *cost*. An action $o$ is applicable in state $s$ if $pre(o) \subseteq s$. Applying action $o$ in state $s$ results in a state denoted by $s[\![o]\!]$ where $s[\![o]\!][v] = eff(o)[v]$ for all $v \in vars(eff)$ and $= s[\![o]\!][v] = s[v]$ for all other variables. An action sequence $\pi = \langle o_1, \cdots, o_n \rangle$ is applicable in state $s$ if there are states $s_0, \cdots, s_n$ such that $o_i$ is applicable in $s_{i-1}$ and $s_{i-1}[\![o_i]\!] = s_i$ for $0 \leq i \leq n$. We denote $s_n$ by $s[\![\pi]\!]$. For convenience we often write $o_1, \cdots, o_n$ instead of $\langle o_1, \cdots, o_n \rangle$. An action sequence with $s_0[\![\pi]\!] \in \mathcal{S}_{s_\star}$ is called a *plan*. The cost of a plan $\pi$, denoted by $cost(\pi)$ is the summed cost of the actions in the plan. For a planning task $\Pi = \langle \mathcal{V}, \mathcal{O}, s_0, s_\star \rangle$, the set of all plans is denoted by $\mathcal{P}_\Pi$. A plan $\pi$ is *optimal* if its cost is minimal among all plans in $\mathcal{P}_\Pi$. For a plan $\pi$, we denote by $MS(\pi)$ the multiset[1] of actions in $\pi$. Note that two different plans $\pi$ and $\pi'$ can have $MS(\pi) = MS(\pi')$. We call such plans *reordering* of each other. Reorderings of actions of a plan that are plans are called *valid* reorderings.

In this paper, we will use a logistics task, depicted in Figure 1, as our running example. This task has two cities, with two locations each, L$_1$ and L$_2$, three trucks, T$_1$ (left), and T$_2$, T$_3$ (right), that can drive within their cities, one airplane, A, that can fly between the airport locations, Apt$_1$ and Apt$_2$, and four packages, P$_1$ to P$_4$, that need to be transported from their initial locations to some specified goal locations. The initial and goal locations of all objects are shown in Figure 1 and marked with dashed arrows. Assuming all actions hav-

---

[1] A set with possible multiple occurences of the same element.

ing unit cost, a cost-optimal plan for this task will consists of 20 actions. Example plans $\pi_a$, $\pi_b$, and $\pi_c$ are depicted in Figure 2.

Given a plan $\pi$, it is sometimes possible to obtain a different plan of equivalent cost without solving the planning task again. Two of such possible ways: action reordering and deriving symmetric plans, are exploited by state-of-the-art top-$k$ planners (Katz et al. 2018). While action reordering is performed using search and may be time consuming, symmetric plans can be obtained using structural symmetries (Shleyfman et al. 2015). Structural symmetries are permutations of variable values and actions that induce automorphisms of the state transition graph. Here, we present the definition of structural symmetries for $\mathrm{SAS}^+$ as was given by Sievers et al. (2017).

**Definition 1 (structural symmetry)** *For a* $\mathrm{SAS}^+$ *planning task* $\Pi = \langle \mathcal{V}, \mathcal{O}, s_0, s_\star \rangle$, *let* $F$ *be the set of* $\Pi$'s *facts, i. e. pairs* $\langle v, d \rangle$ *with* $v \in \mathcal{V}$, $d \in dom(v)$. *A structural symmetry for* $\Pi$ *is a permutation* $\sigma : \mathcal{V} \cup F \cup \mathcal{O} \to \mathcal{V} \cup F \cup \mathcal{O}$, *where:*

1. $\sigma(\mathcal{V}) = \mathcal{V}$ *and* $\sigma(F) = F$ *such that* $\sigma(\langle v, d \rangle) = \langle v', d' \rangle$ *implies* $v' = \sigma(v)$;
2. $\sigma(\mathcal{O}) = \mathcal{O}$ *such that for* $o \in \mathcal{O}$, $\sigma(pre(o)) = pre(\sigma(o))$, $\sigma(\textit{eff}(o)) = \textit{eff}(\sigma(o))$, $cost(\sigma(o)) = cost(o)$;
3. $\sigma(s_\star) = s_\star$;

*where* $\sigma(\{x_1, \ldots, x_n\}) := \{\sigma(x_1), \ldots, \sigma(x_n)\}$, *and* $s' := \sigma(s)$ *is the partial state obtained from the partial state* $s$ *s.t. for all* $v \in vars(s)$, $\sigma(\langle v, s[v] \rangle) = \langle v', d' \rangle$ *implies* $s'[v'] = d'$.

A structural symmetry $\sigma$ *stabilizes* the state $s$ if $\sigma(s) = s$. Given a plan $\pi = o_1 \ldots o_n$ and a structural symmetry $\sigma$ that stabilizes the initial state, applying the permutation $\sigma$ to each action in the plan results in a necessarily valid plan $\sigma(\pi) = \sigma(o_1) \ldots \sigma(o_n)$ of the same cost. Note that $\sigma(\pi)$ is not a reordering of $\pi$, since $\sigma$ may map actions from $\pi$ to actions outside of $\pi$.

In our example, the structural symmetries can detect symmetries between two of the trucks $T_2$ and $T_3$, between the two packages that are initially in $L_1$, and between the two packages that are initially in $L_2$. Thus, structural symmetries can be used to obtain additional plans from a given plan. In our example, the plan $\pi_c$ in Figure 2 can be obtained from $\pi_a$ using the symmetry between the trucks $T_2$ and $T_3$. Note that these two plans use different actions and thus are not reorderings of each other. The plan $\pi_b$, on the other hand, is a reordering of $\pi_a$, changing the order between the first two actions. These two plans are not symmetric, since mapping the action (load $P_4$ $T_2$ $L_2$) to (load $P_3$ $T_2$ $L_2$) would also require mapping (unload $P_4$ $T_2$ $L_2$) to (unload $P_3$ $T_2$ $L_2$). Naturally, there exist pairs of plans that are both symmetric and reordering of each other. There are 6602112 cost-optimal plans in our example, half of them are reorderings of the plan $\pi_a$ and the other half are reordering of $\pi_c$.

Lastly, the *top-$k$ planning problem* (Sohrabi et al. 2016; Katz et al. 2018) is defined as follows.

**Definition 2 (top-$k$ planning problem)** *Given a planning task* $\Pi = \langle \mathcal{V}, \mathcal{O}, s_0, s_\star \rangle$ *and a natural number $k$, find a set of plans* $P \subseteq \mathcal{P}_\Pi$ *such that:*

*(i) for all plans* $\pi \in P$, *if there exists a plan* $\pi'$ *for* $\Pi$ *with* $cost(\pi') < cost(\pi)$, *then* $\pi' \in P$, *and*

*(ii)* $|P| \leq k$, *with* $|P| < k$ *implying* $P = \mathcal{P}_\Pi$.

*An instance of the top-$k$ planning problem* $\langle \Pi, k \rangle$, *is called solvable if* $|P| = k$ *and unsolvable if* $|P| < k$.

The objective of top-$k$ planning is finding $k$ plans of lowest costs for a planning task $\Pi$ and thus optimal planning is the special case of top-1 planning.

## 3 Top-quality Planning

We start by formally defining the top-quality planning problem as the problem of finding all plans of bounded quality.

**Definition 3 (top-quality planning problem)**
*Given a planning task* $\Pi = \langle \mathcal{V}, \mathcal{O}, s_0, s_\star \rangle$ *and a natural number $q$, find the set of plans* $P = \{\pi \in \mathcal{P}_\Pi \mid cost(\pi) \leq q\}$.

The top-quality planning problem is well-defined and always has a solution. Note that one can exploit existing tools for top-k planning to derive solutions to the top-quality planning problem, by setting $k$ to a large value and adding another stopping criteria, once a plan $\pi$ of $cost(\pi) > q$ was obtained. In such cases, $P$ would explicitly contain all plans with $cost(\pi) \leq q$. This was done by Vadlamudi and Kambhampati (2016) as the first step in their algorithm, although they do not formally define the top-quality problem. These explicit sets of plans can get prohibitively large. Further, some of the plans in that set, although different as sequences of actions, could be considered equivalent from the underlying application perspective. If, in addition, it would be possible to escape the need for generating all these equivalent plans, the performance of the planners could improve significantly.

Let $N$ be some equivalence relation on the set of plans $\mathcal{P}_\Pi$. For a plan $\pi \in \mathcal{P}_\Pi$, by $N[\pi]$ we denote the equivalence class of $\pi$, which is a set of all plans that are equivalent to $\pi$ under $N$. Slightly abusing the notation, for a set of plans $P$, by $N[P]$ we denote the union of the equivalence classes $\bigcap_{\pi \in P} N[\pi]$. Using that equivalence relation, we can define the quotient top-quality problem as follows.

**Definition 4 (quotient top-quality planning problem)**
*Given a planning task* $\Pi = \langle \mathcal{V}, \mathcal{O}, s_0, s_\star \rangle$, *an equivalence relation $N$ over its set of plans* $\mathcal{P}_\Pi$, *and a natural number $q$, find a set of plans* $P \subseteq \mathcal{P}_\Pi$ *such that* $\bigcup_{\pi \in P} N[\pi]$ *is the solution to the top-quality planning problem.*

For equivalence relations that preserve plan cost the quotient top-quality planning problem always has a solution. Note that solutions to top-quality planning are solutions to the quotient top-quality planning under the identity equivalence relation. Further, while there is one possible solution to the top-quality planning problem, there can be many solutions to a quotient top-quality problem, defined by representatives of each equivalence class. Further, nothing in our definition prevents a solution from including more than one plan per equivalence class, the only restriction is that all equivalence classes have to be represented.

In this paper, we focus on one specific equivalence relation, considering two plans to be equivalent if their action multi-sets are. Formally, we consider the equivalence relation

$$\mathrm{U}_\Pi = \{(\pi, \pi') \mid \pi, \pi' \in \mathcal{P}_\Pi, \mathrm{MS}(\pi) = \mathrm{MS}(\pi')\}.$$

Thus, the main computational problem we consider in this paper is as follows.

**Definition 5 (unordered top-quality planning problem)**
*Given a planning task* $\Pi = \langle \mathcal{V}, \mathcal{O}, s_0, s_\star \rangle$ *and a natural number* $q$, *find a set of plans* $P \subseteq \mathcal{P}_\Pi$ *such that* $P$ *is a solution to the quotient top-quality planning problem under the equivalence relation* $\mathrm{U}_\Pi$.

Note that while the solution to the top-quality planning problem can be obtained from a solution to the unordered top-quality planning problem, using a simple algorithm that generates all possible valid reordering for each plan in the solution, this is not the focus of current work. Focusing on the unordered top-quality planning problem allows us to generate reorderings of the same plan only if and when these reorderings are actually needed.

## 4 Computation of Top-quality Plans

In order to obtain a solver for the computational problem specified above, we take an approach similar to Katz et al. (2018), and iteratively generate plans using an existing cost-optimal planner, and construct planning tasks with a reduced set of plans, by forbidding exactly the plans found so far. In contrast to Katz et al. (2018), we forbid not only a specific plan, but also all its possible reorderings. In order to achieve that, we thus instead of forbidding plans as sequences of actions, forbid plans as multi-sets. To be able to do that, we need to come up with a reformulation of a planning task that forbids all plans with the exact number of appearances for each action. Similar reformulation was recently suggested by Katz and Sohrabi (2019) for diverse planning. The reformulation can forbid a single multi-set, and thus for a set of plans, the union of their multi-sets was forbidden in each consecutive iteration. That way, possibly additional plans were forbidden. For diverse planning, that did not pose a problem. In our case, however, we need to ensure that we forbid *exactly* the set of plans that were previously found. For that, in what follows, we adapt the reformulation of Katz and Sohrabi (2019) accordingly.

Alternatively, the reformulation of Katz and Sohrabi (2019) can be used directly, creating a sequence of planning tasks, similarly to the way it was done in top-$k$ planning (Katz et al. 2018). This. however, poses two problems: the reformulated planning task size grows fast with each iteration, and, as in the iterative top-$k$ planner, the mapping between the reformulated and original actions must be constantly maintained.

In this work, at each iteration we reformulate the original planning task to forbid all plans found so far. In this case, we do not need to maintain the mapping between the reformulated and original actions and keep the reformulated task size smaller. In the rest of this section we adapt the definition of Katz and Sohrabi (2019) to a set of plans (as multi-sets), present an algorithm that exploits the adapted definition to derive top-quality solutions, and prove its soundness and completeness. We start by presenting the definition.

### 4.1 Forbidding a Plan as a Multi-set of Actions

Slightly simplifying the definition of Katz and Sohrabi (2019), we present a task reformulation that ignores orders between actions in a plan and thus also forbids all possible reorderings of a given plan, as well as all sub-plans.

**Definition 6** *Let* $\langle \mathcal{V}, \mathcal{O}, s_0, s_\star \rangle$ *be a planning task and* $\pi$ *be a plan. The task* $\Pi_\pi^- = \langle \mathcal{V}', \mathcal{O}', s_0', s_\star' \rangle$ *is defined as follows.*

- $\mathcal{V}' = \mathcal{V} \cup \{\overline{v}\} \cup \{\overline{v}_o \mid o \in \pi\}$, *with* $\overline{v}$ *being a binary variable, and* $dom(\overline{v}_o) = \{0, \ldots, m_o\}$, *where* $m_o$ *is the number of occurences of* $o$ *in* $\pi$,

- $\mathcal{O}' = \{o^e \mid o \in \mathcal{O} \setminus \pi\} \cup \bigcup_{i=0}^{m_o}\{o_i^f \mid o \in \pi\}$, *where*
  $pre(o^e) = pre(o), \quad eff(o^e) = eff(o) \cup \{\langle \overline{v}, 0 \rangle\},$
  $pre(o_i^f) = pre(o) \cup \{\langle \overline{v}_o, i \rangle\},$
  *for* $0 \leq i < m_o$, $eff(o_i^f) = eff(o) \cup \{\langle \overline{v}_o, i{+}1 \rangle\}$,
  $eff(o_{m_o}^f) = eff(o) \cup \{\langle \overline{v}, 0 \rangle\}$, *and*
  $cost'(o^e) = cost'(o_i^f) = cost(o), 0 \leq i \leq m_o$,

- $s_0'[v] = s_0[v]$ *for all* $v \in \mathcal{V}$, $s_0'[\overline{v}] = 1$, *and* $s_0'[\overline{v}_o] = 0$ *for all* $o \in \pi$, *and*

- $s_\star'[v] = s_\star[v]$ *for all* $v \in \mathcal{V}$ *s.t.* $s_\star[v]$ *defined, and* $s_\star'[\overline{v}] = 0$.

The semantics of the reformulation is as follows. The variable $\overline{v}$ starts from the value 1 and switches to 0 when an action is applied that is not from plan $\pi$ treated as a multi-set. Once a value 0 is reached indicating a deviation from plan $\pi$, it cannot be switched back to 1. The finite-domain variables $\overline{v}_o$ encode the number of applications of the action $o$. The actions $o_i^f$ are copies of the action $o$ in $\pi$, counting the number of applications of $o$, as long as the number is not higher than the number of times it appears in $\pi$. Once the number of applications exceeds $m_o$, $\overline{v}$ is set to 0.

### 4.2 Forbidding Multiple Plans Exactly

In order to forbid multiple plans, the greedy approach of Katz and Sohrabi (2019) forbids the super-set of these plans by taking a super-set of the multi-sets representing the plans. In our case, when optimality is required, we cannot follow the same approach. Instead, we present a reformulation that forbids exactly these plans and their sub-plans, and the possible reorderings. Our reformulation extends the one in Definition 6, by introducing a binary variable for each plan, encoding whether the plan was deviated from.

**Definition 7** *Let* $\langle \mathcal{V}, \mathcal{O}, s_0, s_\star \rangle$ *be a planning task,* $P$ *be a set of plans, and* $\mathcal{O}_P = \{o \mid o \in \pi, \pi \in P\}$. *The task* $\Pi_P^- = \langle \mathcal{V}', \mathcal{O}', s_0', s_\star' \rangle$ *is defined as follows.*

- $\mathcal{V}' = \mathcal{V} \cup \{\overline{v}_\pi \mid \pi \in P\} \cup \{\overline{v}_o \mid o \in \mathcal{O}_P\}$, *with* $\overline{v}_\pi$ *being binary variables, and* $dom(\overline{v}_o) = \{0, \ldots, m_o\}$, *where* $m_o = \max_{\pi \in P}\{m_o^\pi\}$ *and* $m_o^\pi$ *is the number of occurrences of* $o$ *in* $\pi$,

- $\mathcal{O}' = \{o^e \mid o \in \mathcal{O} \setminus \mathcal{O}_P\} \cup \{o_i^f \mid o \in \mathcal{O}_P, 0 \le i \le m_o\}$, *where*

$$pre(o^e) = pre(o), \, eff(o^e) = eff(o) \cup \{\langle \overline{v}_\pi, 0 \rangle \mid \pi \in P\},$$
$$pre(o_i^f) = pre(o) \cup \{\langle \overline{v}_o, i \rangle\},$$
$$eff(o_i^f) = eff(o) \cup \{\langle \overline{v}_o, i{+}1 \rangle\} \cup \{\langle \overline{v}_\pi, 0 \rangle \mid i = m_{op}^\pi\}$$
$$\text{for } 0 \le i < m_o,$$
$$eff(o_{m_o}^f) = eff(o) \cup \{\langle \overline{v}_\pi, 0 \rangle \mid \pi \in P\}, \, and$$
$$cost'(o^e) = cost'(o_i^f) = cost(o), \, 0 \le i \le m_o,$$

- $s_0'[v] = s_0[v]$ *for all* $v \in \mathcal{V}$, $s_0'[\overline{v}_\pi] = 1$ *for all* $\pi \in P$, *and* $s_0'[\overline{v}_o] = 0$ *for all* $o \in \mathcal{O}_P$, *and*
- $s_\star'[v] = s_\star[v]$ *for all* $v \in \mathcal{V}$ *s.t.* $s_\star[v]$ *defined, and* $s_\star'[\overline{v}_\pi] = 0$ *for all* $\pi \in P$.

## 4.3 Using the Reformulation

Algorithm 1 exploits the reformulation in Definition 7 to find a solution to the unordered top-quality planning problem. The algorithm incrementally finds the set $P$ of top quality plans. Starting with the empty set $P = \emptyset$ and assuming $\Pi_\emptyset^- = \Pi$, we use an optimal planner iteratively to find an optimal plan $\pi$ to the planning task $\Pi_P^-$. Once a plan is found, it is added to the set of found plans $P$. Then, the new reformulation $\Pi_P^-$ is constructed from $\Pi$ for the next iteration. The algorithm stops when a plan $\pi$ is generated such that $cost(\pi) > q$. Note that the algorithm results in a set $P$ of sequential plans, with no two plans being reorderings of each other. Similarly to Katz et al. (2018), at each iteration, after the plan $\pi$ was found, we use structural symmetries to generate from $\pi$ additional plans that are symmetric (Shleyfman et al. 2015) to $\pi$, and add these that are not reorderings of $\pi$ to the set $P$. Finally, since the first step results in an optimal plan, the quality can be defined relatively to the cost of the optimal plan rather than an absolute number.

**Theorem 1** *Algorithm 1 is sound and complete for unordered top-quality planning when using cost-optimal planners that find shortest (in the number of actions) cost-optimal plans.*

**Proof:** Let $P$ be the set of plans returned by Algorithm 1 and let $\pi_f$ be the plan found when the algorithm breaks. Since $\pi_f$ is an optimal plan to $\Pi_P^-$ and $cost(\pi_f) > q$, we need to show that $\Pi_P^-$ forbids exactly the plans in $U_\Pi[P]$. For a plan $\pi \in P$, $\Pi_P^-$ has a variable $\overline{v}_\pi$ that reaches its goal value only when the number of applications of some action exceeds the number of appearances of that action in $\pi$. Thus, $\pi$ is not a plan for $\Pi_P^-$. Since Definition 7 treats plans as multi-sets, this is true also for all $\pi' \in U_\Pi[\pi]$.

Let $P_1, \dots, P_n$ denote the sets of plans at the beginning of each algorithm iteration and let $\pi_1, \dots, \pi_n = \pi_f$ be the optimal plans found by the algorithm in these iteration, with $\pi_i$ being an optimal plan to $\Pi_{P_i}^-$. Let $\pi$ be a plan for $\Pi$ such that $cost(\pi) \le q$. If $\pi \notin U_\Pi[P]$, there exists $k$ such that $\pi$ is a plan for $\Pi_{P_k}^-$, but not for $\Pi_{P_{k+1}}^-$. Let $P' = P_{k+1} \setminus P_k$ be the plans forbidden in $\Pi_{P_{k+1}}^-$ but not in $\Pi_{P_k}^-$. Then, there exists $\pi' \in P'$ such that $MS(\pi) \subseteq MS(\pi')$. If $MS(\pi) = MS(\pi')$, then $\pi \in P$ and we are done. Assume that $MS(\pi)$

**Algorithm 1** Iterative unordered top-quality planning.

**Input:** Planning task $\Pi$, quality bound $q$
  $P \leftarrow \emptyset$
  $\Pi' \leftarrow \Pi$
  **while** True **do**
    $\pi \leftarrow$ optimal plan to $\Pi'$
    **if** $cost(\pi) > q$ **then**
      **break**
    **end if**
    $P \leftarrow P \cup \{\pi\} \cup \{\pi' \mid \pi' \text{ is symmetric to } \pi, \pi' \notin U_\Pi[\pi]\}$
    $\Pi' \leftarrow \Pi_P^-$ according to Definition 7
  **end while**
  **return** $P$

is a proper subset of $MS(\pi')$. Note that $\pi'$ is a reordering of a plan that is symmetric to $\pi_k$, which was the optimal plan found for $\Pi_{P_k}^-$. Assuming that our optimal planner finds shorter optimal plans before longer ones, a plan $\pi$ for $\Pi_{P_k}^-$ would be found before $\pi_k$, contradicting the assumption that $MS(\pi)$ is a proper subset of $MS(\pi')$. $\qquad\square$

## 5 Experimental Evaluation

In order to evaluate the feasibility of our suggested approach for unordered top-quality planning, we have implemented our approach as part of the ForbidIterative planners collection (Katz, Sohrabi, and Udrea 2019), which is implemented on top of the Fast Downward planning system (Helmert 2006). The collection, among other, includes the implementation of the iterative top-k planner (Katz et al. 2018). The experiments were performed on Intel(R) Xeon(R) CPU E7-8837 @2.67GHz machines, with the time and memory limit of 30min and 2GB, respectively. The benchmark set consists of all STRIPS benchmarks from optimal tracks of International Planning Competitions (IPC) 1998-2018, a total of 1797 tasks in 64 domains. Our baseline for the comparison is a simple approach, using a top-k planner with a large k value, $10^9$, stopping if a plan of quality above the bound was reached. We use *NaiveS*, the best perfoming configuration of the iterative top-k planner (Katz et al. 2018), that exploits both symmetries and plan reorderings. The purpose of setting k to a large number is to allow the top-k planner to exploit the entire 30min time interval. Among tasks solved, the largest number of plans found by the top-k planner was 60480. For tasks not solved, the maximal number of plans found by the top-k planner was 767501. Note that reading and writing such large amounts of plans is time consuming by itself. For each task, the quality bound is computed using the cost of the first found (optimal) plan, multiplied by a constant[2]. We experiment with four different quality bound multipliers, namely $q_m = 1.0$ (optimal plans only), 1.05, 1.1, and 1.2 of the optimal plan cost. For larger quality bounds, both approaches had low coverage, and thus we do not report these results. Note, $q$ can be any natural number as mentioned in Definition 5.

---

[2]This is not an overhead, as at least one optimal planner run needs to be performed anyway.

| Coverage | $q_m\!=\!1.00$ K-tq | tq | $q_m\!=\!1.05$ K-tq | tq | $q_m\!=\!1.10$ K-tq | tq | $q_m\!=\!1.20$ K-tq | tq |
|---|---|---|---|---|---|---|---|---|
| airport | 7 | **21** | 7 | **18** | 6 | **17** | 6 | **17** |
| blocks | 16 | **17** | 16 | **17** | 10 | **13** | 8 | **9** |
| data-network18 | 0 | **1** | 0 | 0 | 0 | 0 | 0 | 0 |
| depot | 2 | 2 | 2 | 2 | 0 | **2** | 0 | **1** |
| driverlog | 5 | **9** | 5 | **9** | 1 | **7** | 1 | **4** |
| floortile11 | 0 | **2** | 0 | **2** | 0 | 0 | 0 | 0 |
| ged14 | 5 | **7** | 5 | **7** | 5 | **7** | 5 | **7** |
| gripper | 1 | **4** | 1 | **3** | 0 | **2** | 0 | **2** |
| logistics00 | 3 | **16** | 3 | **13** | 1 | **10** | 0 | **6** |
| logistics98 | 0 | **4** | 0 | **2** | 0 | **1** | 0 | 0 |
| miconic | 18 | **27** | 18 | **26** | 11 | **16** | 10 | **12** |
| movie | 0 | **1** | 0 | **1** | 0 | **1** | 0 | 0 |
| mprime | 18 | **19** | 18 | **19** | 18 | **19** | 6 | **11** |
| mystery | 20 | 20 | 20 | 20 | 20 | 20 | 13 | **15** |
| nomystery11 | 9 | **13** | 7 | **11** | 4 | **8** | 2 | **5** |
| openstacks08 | 0 | **2** | 0 | **2** | 0 | **2** | 0 | **2** |
| parcprinter08 | 6 | **15** | 5 | **12** | 5 | **12** | 5 | **11** |
| parcprinter11 | 3 | **11** | 2 | **8** | 2 | **8** | 2 | **7** |
| pegsol08 | 21 | **23** | 21 | **23** | 21 | **22** | 8 | **17** |
| pegsol11 | 8 | **13** | 8 | **13** | 8 | **12** | 2 | **5** |
| pipes-notank | 5 | **11** | 5 | **11** | 3 | **7** | 1 | **4** |
| pipes-tank | 2 | **4** | 2 | **4** | 2 | **4** | 1 | 1 |
| psr-small | 37 | **46** | 26 | **40** | 22 | **36** | 16 | **24** |
| rovers | 3 | **6** | 3 | **6** | 2 | **4** | 0 | **3** |
| satellite | 2 | **5** | 2 | **5** | 1 | 1 | 0 | **1** |
| scanalyzer08 | 4 | **5** | 4 | **5** | 4 | 4 | 3 | 3 |
| scanalyzer11 | 1 | **2** | 1 | **2** | 1 | 1 | 1 | 1 |
| spider18 | 5 | 5 | 3 | **4** | 0 | 0 | 0 | 0 |
| storage | 8 | **14** | 8 | **14** | 7 | **11** | 6 | **7** |
| tetris14 | 1 | **2** | 1 | **2** | 1 | **2** | 0 | **1** |
| tidybot11 | 5 | **7** | 2 | **5** | 1 | **3** | 1 | 1 |
| tpp | 4 | **6** | 4 | **5** | 2 | **5** | 2 | **5** |
| transport08 | 6 | **7** | 1 | 1 | 1 | 1 | 0 | 0 |
| transport14 | 0 | **1** | 0 | 0 | 0 | 0 | 0 | 0 |
| trucks | 1 | **2** | 1 | **2** | 0 | **1** | 0 | 0 |
| visitall11 | 8 | 8 | 7 | 7 | 5 | **6** | 5 | 5 |
| woodwork08 | 3 | **8** | 2 | **6** | 1 | **4** | 0 | **2** |
| woodwork11 | 0 | **3** | 0 | **1** | 0 | 0 | 0 | 0 |
| zenotravel | 7 | 7 | 7 | 7 | 4 | **5** | 3 | **4** |
| Sum other | 25 | 25 | 23 | 23 | 21 | 21 | 18 | 18 |
| Sum (1797) | 269 | **401** | 240 | **358** | 190 | **295** | 125 | **211** |

Table 1: The coverage results comparing to top quality planning via top-k planning, for various quality bounds.

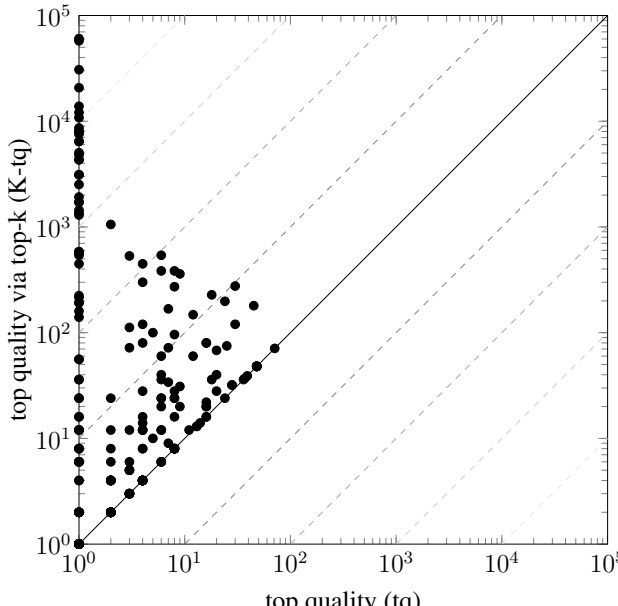

Figure 3: Per-task comparison of the solution encoding size.

Table 1 depicts the per-domain summed coverage, comparing our technique, tq, to the baseline, K-tq, for four quality bound multipliers. Each task gets a coverage of 1 if and only if the planner proved there is no other plan within quality bound, by either finding a plan above the bound or proving there are no other plans. Note first that out of the 64 domains, there are 18 domains where all optimal plans could not be found for any tasks, with any approach. There are 7 more domains where there is no difference in coverage between the baseline and our approach, for all tested quality bounds. These 25 domains are summarized in the *Sum other* row of Table 1. Out of the remaining 39 domains, the coverage never gets worse and it gets better (often significantly better) for at least one of the tested quality bounds. Extreme examples are AIRPORT, LOGISTICS00, and PSR-SMALL where the increase in coverage for some quality

bounds is by 10 instances or more. Overall, there is a clear benefit of the suggested approach over the baseline.

Another benefit of our approach is a compact representation of the solution. Figure 3 shows a per-task comparison of the number of plans in the solution for each of the approaches, for the quality bound multiplier $q_m = 1.0$, for tasks solved by both approaches. First, out of the total of 263 such tasks, there are 111 tasks on the diagonal. Out of the remaining 152 tasks (all above the diagonal), 73 tasks have a single optimal plan found by our approach, while the baseline needs to find multiple optimal plans, which are all reorderings of the same plan, with the maximal number of 60480 reorderings found. When the number of valid reordering is larger, the baseline approach fails before being able to find all optimal plans.

Finally, Figure 4 compares the reformulated task size of our approach to the baseline one. We compare the last generated task reformulation, for tasks solved by both approaches, for the quality bound multiplier $q_m = 1.0$. The task size is measured here by the number of facts, i.e., variable value pairs. While the larger tasks are not necessarily harder for a classical planner, this is usually the case. Our experiments clearly show that our approach creates tasks of sizes almost two orders of magnitude smaller than the baseline approach.

## 6 Conclusions and Future Work

In this work we have shown a way of obtaining all plans of bounded solution quality, representing plan reorderings implicitly and thus escaping the need for counting plans. We have presented a novel reformulation of a planning task that forbids exactly the set of given plans, their reorderings, and all subplans thereof. We have formally defined the family of computational problems in top-quality planning and

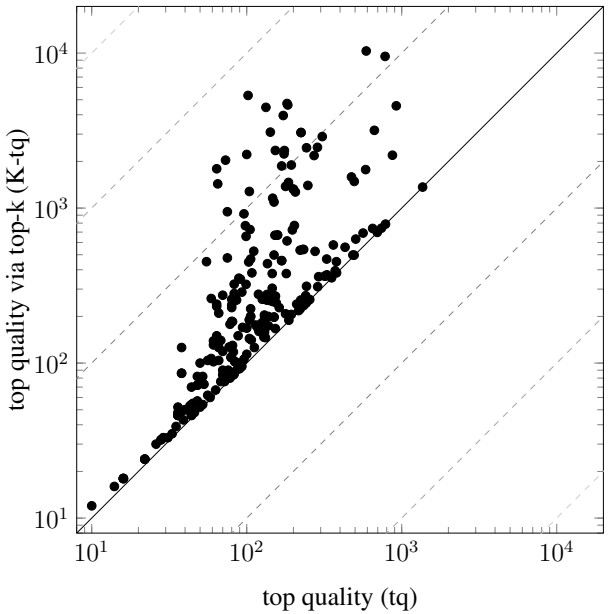

Figure 4: Final constructed task size in terms of the number of facts.

have implemented a first planner for unordered top-quality planning. The planner, exploiting the new reformulation, has empirically shown to perform significantly better than the straightforward approach of exploiting top-$k$ planners with a large bound $k$, as it is often done in practice.

For future work, one promising direction is exploring the use of top-quality instead of top-$k$ planners in planning applications. Another possible direction is creating a top-$k$ planner based on the unordered top-quality planner, exploiting the more compact task representation. Further, (unordered) top-quality planners can be used to obtain solutions to diverse planning, when solution cost is also considered (Vadlamudi and Kambhampati 2016).

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
