# OpenReview forum: "Top-Quality: Finding Practically Useful Sets of Best Plans"
_icaps-conference.org/ICAPS/2019/Workshop/HSDIP_

### Official Review · AnonReviewer1 · 2019-03-29
**The paper formally defines the problem of finding all plans up to a certain solution cost, introduces a planner for this problem and shows that it outperforms an approach using a top-k planner.**

**Rating:** 9
**Confidence:** 4

**Review:**

The paper is written well, makes interesting theoretical and practical contributions and is therefore a nice fit for the workshop. I only have minor comments:

I think it would be good to name the optimal planner you use for the experiments.

I think Figure 3 would benefit from an explanation. Why is the solution encoding size so much bigger for K-tq?

I think Figure 4 shows a difference of almost 2 orders of magnitude, not "several" orders of magnitude.

The paper doesn't promise to make any code or results available. In order to allow others to build on this work, I would encourage the authors to share the code and the empirical results.

Typos:
* Avoid nesting brackets for "e.g." citations.
* a planning task--s--
* only a handful of work++s++ take the plan quality --under-- ++into++ consideration
* harder for ++a++ classical planner

---

> ### Author Response · Authors · 2019-04-09
> **We thank the reviewers for their constructive feedback.**
>
> Indeed, the optimal planner used was not specified. The planner is using OSS with LM-cut heuristic.
>
> The size of the encoding for K-tq is indeed much larger. The reformulation by Katz et al. 2018 explicitly forbids sequences of actions, while our suggested reformulation is much more compact, forbidding plans as multisets.
>
> We are making efforts toward having the code publicly available as soon as possible.

---

### Official Review · AnonReviewer2 · 2019-04-05
**Interesting new problem introduced + good first algorithm to solve it**

**Rating:** 8
**Confidence:** 4

**Review:**

This paper introduces a novel approach to finding possibly large sets
of diverse plans to a planning task with quality that is still within
a given ratio from the optimal cost. The authors baptize their
approach top-quality planning, finding all plans up to a fixed cost q
and representing these plans is a very compact form.

The algorithm proposed by the paper works by repeatedly solving a
planning task with an optimal planner, by, after each iteration,
forbidding all permutations of the last found plan for later
iterations. The latter is done by reformulating the planning task,
essentially counting the number of actions occurrences.

The paper introduces an interesting new problem and a nice (and
well-working) solution to it. The theory is sound and the experimental
evaluation shows that the proposed algorithm works well (compared to
known approaches to somewhat different problems that can be used to
solve the top-quality planning problem). This makes it a good
contribution to the workshop.

- Why do you restrict the constant q to natural numbers? Why would it
not work for (positive) reals?

Minor things:
- abstract: "introduced additional*ly* restrictions"
- please remove the copyright statement from the first page
- "(e.g., (Katz et al. 2018)..)" should probably be "(e.g., Katz et al. (2018)..)"
- plans should probably be defined as starting from the initial state.
alternatively, you could define a plan for a task as starting in its
initial state.
- you should mention that the blue dashed lines in figure 1 indicate
the goals
- in the definition of structural symmetries, and also after that in
the paper, you mix notation for the domain of a variable v ({\cal D}(v) vs dom(v))
and also the variables of an assignment s ({\cal V}(s) vs vars(s))
- a symmetry that "stabilizes the initial state" is not defined
- you should show the indices of the trucks in figure 1, it is not
clear that T_2 and T_3 are the ones starting in the right
- the function "reord()" is not defined
- the bibliography does not seem to be in AAAI style

---

> ### Author Response · Authors · 2019-04-09
> **We thank the reviewers for their constructive feedback.**
>
> We will fix the minor issues for the final version.
>
> The restriction of q to natural numbers is due to the (typical) restriction of action costs to natural numbers. There is nothing that requires q to be a natural number and if the action costs are reals, then q should be real as well.

---

> > ### Comment · AnonReviewer2 · 2019-04-10
> > **Thank you for the clarification**
> >
> > Since many papers use positive reals as action cost, I was wondering why you make this restriction. I didn't check, though, that you also "only" use natural numbers for action cost.

---

### Public Comment · ~Patrik_Haslum1 · 2019-04-09
**overlap with "Reshaping Diverse Planning"?**

This paper has some overlap with another submission ("Reshaping Diverse Planning: Let There Be Light").
The two papers define different problems: here, finding all solutions within a factor of optimal cost; in the other paper, finding a collection of K plans with bounds on cost and/or some diversity metric. However, the approach and the specific reformulation used in both papers is essentially the same.
Given that we have (perhaps) slightly more submissions than we can accept into a 1-day workshop, in the interest of composing a more diverse as well as high-quality workshop program,  IF it happens to be the case that there is a substantial overlap in the authorship of these two papers, could we propose to the authors that they reformulate both problems and the common approach into one paper? (One may call it a "merge and shrink".)

---

> ### Comment · Program_Chairs · 2019-04-09
> **Good point**
>
> I think this is a reasonable suggestion, if the change doesn't require extensive changes to the content of the papers having to add some "bridging" background or related work . Since we are not charging for extra pages in the "proceedings", I don't think there is a problem if the paper goes over the page limit (within reason, e.g. 9 pages + refs).
>
> *If* these papers are accepted, then we will most definitely bundle them into the same presentation slot.

---

> ### Author Response · Authors · 2019-04-09
> **re: overlap**
>
> Although there is indeed some overlap with "Reshaping Diverse Planning: Let There Be Light", and some of the authors of the other paper, there are substantial differences between the problems tackled.
>
> Further, the reformulations used are not the same, with the reformulation presented in this paper ensuring that no plans will be missed, and the reformulation in "Reshaping Diverse Planning: Let There Be Light" is more permissive.
>
> Looking at previous years and the current reviews, it seems like there is ample space to accept both submissions. If, however, that is not the case, we might suggest we make a single presentation for both papers and save some presentation time.
>
> Making a new paper out of the two would be extremely time consuming, even the merging part only and the outcome might be somewhat confusing to the reader. Further, it is not clear to us what is the benefit of having the two as a single paper.

---

### Meta-Review · Program_Chairs · 2019-04-25

**Recommendation:** Accept
**Confidence:** 5

**Metareview:**

Dear Authors,
thank you very much for your submission. We are happy to inform you that
we have decided to accept it and we look forward to your talk in the workshop.
Please, go over the feedback in the reviews and correct or update your papers
in time for the camera ready date (May 24).
Best regards
HSDIP organizers